# Self-Evolution Learning for Mixup: Enhance Data Augmentation on Few-Shot Text Classification Tasks

**Haoqi Zheng**[1][*], **Qihuang Zhong**[2][*], **Liang Ding**[3], **Zhiliang Tian**[1][†],
**Xin Niu**[1][†], **Changjian Wang**[1] , **Dongsheng Li**[1], **Dacheng Tao**[4]

[1]College of Computer, National University of Defense Technology
[2]School of Computer Science, Wuhan University  [3]JD Explore Academy  [4]University of Sydney

## Abstract

Text classification tasks often encounter few-shot scenarios with limited labeled data, and addressing data scarcity is crucial. Data augmentation with mixup merges sample pairs to generate new pseudos, which can relieve the data deficiency issue in text classification. However, the quality of pseudo-samples generated by mixup exhibits significant variations. Most of the mixup methods fail to consider the varying degree of learning difficulty in different stages of training. And mixup generates new samples with one-hot labels, which encourages the model to produce a high prediction score for the correct class that is much larger than other classes, resulting in the model's over-confidence. In this paper, we propose a self-evolution learning (SE) based mixup approach for data augmentation in text classification, which can generate more adaptive and model-friendly pseudo samples for the model training. SE caters to the growth of the model learning ability and adapts to the ability when generating training samples. To alleviate the model over-confidence, we introduce an instance-specific label smoothing regularization approach, which linearly interpolates the model's output and one-hot labels of the original samples to generate new soft labels for label mixing up. Through experimental analysis, experiments show that our SE brings consistent and significant improvements upon different mixup methods. In-depth analyses demonstrate that SE enhances the model's generalization ability.

## 1 Introduction

Recently, generative large language models (LLMs) have won great popularity in natural language processing (NLP), and have achieved impressive performance on various NLP tasks (Kocoń

et al., 2023; Peng et al., 2023; Lu et al., 2023c). However, empirical studies (Zhong et al., 2023) suggest that LLMs do not always outperform BERT in some language understanding tasks. Hence, employing BERT is still a viable option in some applications. Text classification tasks often encounter few shot scenarios (e.g. NLI and Paraphrase tasks), where there are limited suitable labeled data available for training. Data augmentation (DA) generates new data by changing the original data through various methods, which enlarges the training dataset to alleviate the issue of data scarcity.

In text classification tasks, DA methods can be divided into two categories: DA methods like EDA (Wei and Zou, 2019), Back-Translation (Kobayashi, 2018), and others based on synthesis such as mixup. The first category conducts DA by altering only the inputs. These methods only alter the inputs to generate new data while maintaining the original labels. These methods are easy to implement, but the input only changes a little thus leading to augmented inputs with limited diversity, which may reduce model generalization. The second category of DA methods modify both inputs and labels, which changes the input samples in a certain way and simultaneously changes the corresponding labels to compose a new sample. These methods tend to generate samples more distinct from the original samples.

Mixup is a DA method that modifies both inputs and labels. It mixes up inputs of samples and their labels, where labels are commonly represented with one-hot encoding. Most of these methods mix up inputs of two samples on their input text (Yun et al., 2019) or hidden-level representations (Verma et al., 2019). However, the pseudo sample, simply combined with two samples, may not be adaptive to the model's learning ability and friendly to the model training. Recently, some work (Sawhney et al., 2022; Park and Caragea, 2022)

---

[*]Haoqi Zheng and Qihuang Zhong contribute equally to this work.
[†]Corresponding Authors

have focused on selecting similar sample pairs for the mixup. Sawhney et al. (2022) select samples according to the embedding similarity. Park and Caragea (2022) merge one sample considering the confidence of the model's predictions. Moreover, in few-shot scenarios, using hard labels (one-hot labels) can lead to issues, where the one-hot labels fail to provide uncertainty of inputs since all the probability mass is given to one class. This results in over-confident models since the largest logit becomes larger than the others which removes the uncertainty of label space (Szegedy et al., 2016). The current label smoothing techniques generate soft labels that cannot dynamically adapt to the model's increasing ability as the training goes on, so they also cannot adjust according to the model's performance at the current stage.

In this paper, we propose self-evolution learning for mixup to achieve data augmentation in text classification tasks. To cater to the model's learning ability, we first divide the training data into easy-to-learn and hard-to-learn subsets. We gradually start from the mixup of easy-to-learn samples and then gradually transition to the mixup operation of hard-to-learn samples. To avoid the model's over-confidence, we introduce an instance-specific label smoothing method, where we linearly interpolate the predicted probability distribution of the original sample and its one-hot label to obtain a soft label. Using this soft label reduces the difference between the model's prediction probability for different classes, which can alleviate the model's over-confidence. Additionally, this instance-specific label can dynamically adapt to the growth of the model's increasing ability and can be customized to the model's current performance. Our method has empirically proven that mixing up in the order of increased difficulty can make the generated samples more adaptive for model training compared to randomly selected samples.

Our contributions are as follows:

• We propose self-evolution learning (SE) for mixup to consider the learning difficulty of samples for data augmentations on text classification tasks.

• We propose an instance-specific label smoothing approach for regularization which can obtain dynamic and adaptive soft labels to alleviate the model's over-confidence and enhance the model's generalization ability.

• Extensive experiments show that our model significantly and robustly improves the mixup method

on few-shot text classification tasks.

## 2 Related Work

### 2.1 Few-shot Text Classification

Driven by the observation that humans can rapidly adapt existing knowledge to new concepts with limited examples, few-shot learning (Fei-Fei et al., 2006) has recently drawn a lot of attention. Few-shot text classification entails performing classification after training or tuning a model on only a few examples. Several studies (Yu et al., 2018; Bailey and Chopra, 2018; Geng et al., 2020) have explored various approaches for few-shot text classification, which mainly involve the traditional machine learning techniques for selecting the optimal category sub-samples.

More recently, ever since Devlin et al. (2019); Brown et al. (2020) show the impressive performances of pre-trained language models (PLMs) on a variety of NLP tasks, a great deal of works (Wu et al., 2019; Bansal et al., 2020) tend to employ the PLMs to tackle the few-shot text classification problem. One line of work aims to fine-tune the PLMs (mainly for discriminative PLMs, such as BERT (Devlin et al., 2019)) with the few-shot training data. Correspondingly, how to design data augmentation methods to better enrich training data has become the focus of these works. Rather than fine-tuning the PLMs, a separate line of research aims to take full advantage of the emergent few-shot learning ability of larger PLMs, *i.e.*, GPT-3 (Brown et al., 2020) and InstructGPT (Ouyang et al., 2022), and use the few-shot training data as the demonstrations for performing in-context learning process (Lu et al., 2023b,a). Considering the BERT-based PLMs are more suitable for text classification tasks, we follow the former research line and focus on exploring the ability of BERT-based PLMs in the few-shot text classification.

### 2.2 Data Augmentation in NLP

Since the bottleneck in few-shot learning is the lack of data, the performance can be easily improved if we can generate more labeled data. Hence, various NLP data augmentation techniques have been proposed, such as EDA (Wei and Zou, 2019), Back-Translation (Kobayashi, 2018) and CBERT (Wu et al., 2019). These methods show remarkable performance in some specific scenarios, however, they mainly focus on altering the original input, resulting in a lack of diversity in the generated samples.

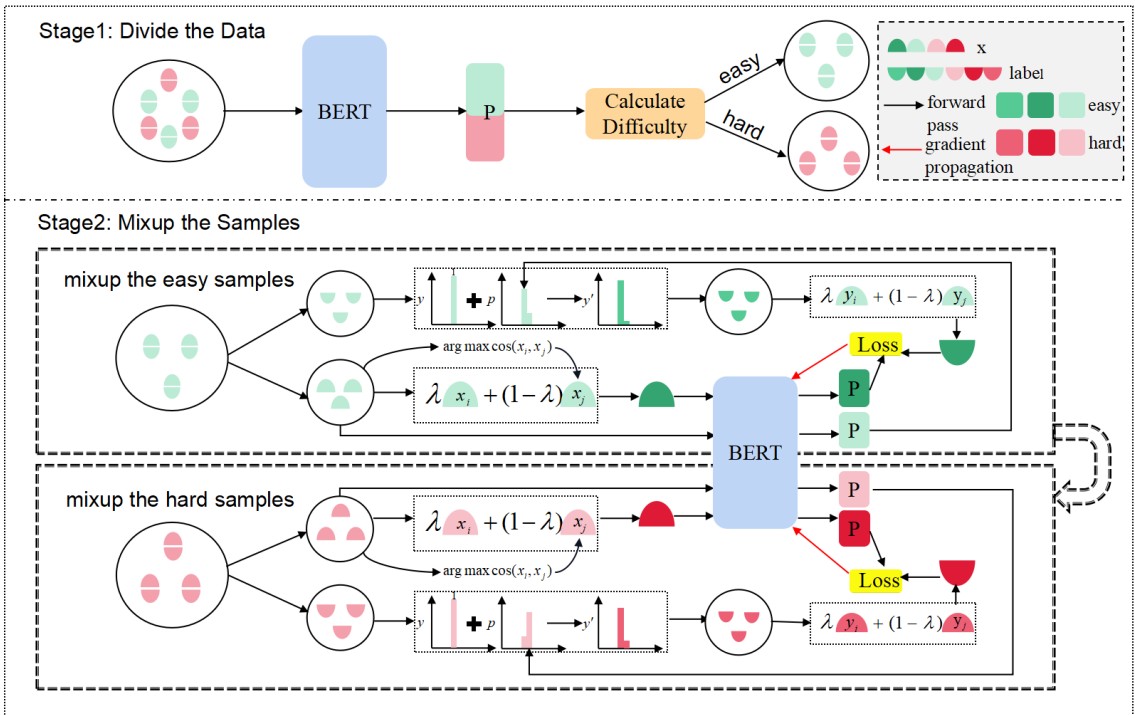

Figure 1: : Overview of our mechanism, which contains two stages: using an existing BERT to divide the data according to difficulty and mixing up the samples to model's training from easy to hard. Best viewed in color.

In response to this problem, Szegedy et al. (2016) first propose a domain-independent data augmentation technique (*i.e.*, mixup) in the computer vision domain, that linearly interpolates image inputs on the pixel-based feature space. Guo et al. (2019) then integrate the mixup with CNN and LSTM for text applications. Furthermore, to achieve better performance, various works (Sun et al., 2020; Cao et al., 2021; Chen et al., 2020; Yoon et al., 2021; Zhang et al., 2022) attempt to improve the mixup technique from two perspectives: 1) how to better merge the two hidden representations, and 2) how to directly perform the mixup on the input sentences.

Although achieving remarkable performance, these previous mixup strategies still have some limitations. Specifically, they (usually) randomly select samples to mix and do not consider the model's learning ability. Some works (Sawhney et al., 2022; Park and Caragea, 2022) have also focused on addressing this issue and proposed various methods for effectively choosing samples. Sawhney et al. (2022) select samples according to the embedding similarity. Park and Caragea (2022) merge one sample considering the confidence of the model's predictions. Along the same research line, in this

paper, we improve the mixup with a more simple-yet-effective self-evolution learning mechanism.

## 3 Method

### 3.1 Overview

For the text classification task in the few-shot scenario, we propose a data augmentation method via mixup, where the training follows an easy-to-hard schedule over the augmented data. First, we construct a text classification model based on the BERT and then employ a mixup method for data augmentation to expand the amount of data (Sec3.2). To make the mixup adaptive for model learning ability, we propose **self-evolution learning** for mixup (Sec3.3). To alleviate the over-confidence problem of the model, we propose an **instance-specific label smoothing regularization method**, which linearly interpolates the model's outputs and one-hot labels of the original samples to generate new soft labels as the label for mixing up (Sec3.4).

### 3.2 Text Classification Model and Mixup

We utilize the BERT (Devlin et al., 2018) for text classification tasks, where the BERT model adopts a multi-layer bidirectional Transformer encoder

architecture and is pre-trained on plain text for masked language modeling.

BERT takes a sequence of words as the input and outputs the representation of the sequence. For text classification tasks, BERT takes the final hidden state $h$ of the first token $[CLS]$ as the sentence representation. Then, we append a softmax function with a linear transformation to generate a probability distribution and the predicted label.

To relieve the data deficiency in few-shot scenarios, we propose a data augmentation method to generate pseudo samples for training the BERT model. The core idea of mixup is to select two labeled data points $(x_i, y_i)$ and $(x_j, y_j)$, where $x$ is the input and $y$ is the label. The algorithm then produces a new sample $(\tilde{x}, \tilde{y})$ through linear interpolation:

$$\tilde{x} = \lambda x_i + (1 - \lambda)x_j \qquad (1)$$
$$\tilde{y} = \lambda y_i + (1 - \lambda)y_j \qquad (2)$$

where $\lambda \in [0, 1]$ denotes the mixing ratio of two samples.

### 3.3 Self-Evolution Learning for Mixup

To make the mixed samples more adaptive and friendly to the model training, we propose a novel mixup training strategy: progressive mixup training from easy to hard. This idea is inspired by human learning behavior: a human's learning schedule usually starts from easier tasks and gradually progresses to more challenging tasks. We first propose the degree of *difficulty* to measure the difficulty of the model in learning samples and then conduct mixup in two stages: (1) dividing the dataset based on the degree of difficulty, and (2) mixup two samples according to the order of difficulty from easy to hard.

To obtain the degree of difficulty $d(x_i)$ for sample $x_i$, we calculate the difference between the model predicted probability on the correct label $p(y_i|x_i)$ and the maximum predicted probability among the wrong labels as Eq.3:

$$d(x_i) = 1 - (p(y_i|x_i) - \max_{y \in C, y \neq y_i} p(y|x_i)), \quad (3)$$

where $y_i$ denotes the ground-truth label, and $C$ denotes the set of all candidate labels.

In the first stage of self-evolution learning (SE), we divide the training data into two datasets according to the degree of difficulty. Given a training set $D$, we calculate the degree of difficulty of each

**Algorithm 1** Mixup with Self-evolution learning and instance-specific label smoothing

**Input**: Labeled set $\mathcal{D}$; Mixup function $\zeta(\cdot)$; Instance-specific label smoothing function $\phi(\cdot)$
**// Stage1: Divide the Data**
**// Calculate Difficulty**
**for** $(x_i, y_i)$ **in** $\mathcal{D}$ **do**
$\quad | \quad d(x_i) = 1 - (p(y_i|x_i) - \max_{y \in C, y \neq y_i} p(y|x_i))$
**end**
$\xi = \mathbf{median}(d(x_i))$
**for** $(x_i, y_i)$ **in** $\mathcal{D}$ **do**
$\quad | \quad \mathcal{D}_{easy} \leftarrow \mathcal{D}_{easy} \cup (x_i, y_i), \quad if \quad d(x_i) \leq \xi$
$\quad | \quad \mathcal{D}_{hard} \leftarrow \mathcal{D}_{hard} \cup (x_i, y_i), \quad if \quad d(x_i) > \xi$
**end**
**// Stage2: Mixup the Samples**
**for** $(x_i, y_i)$ **in** $\mathcal{D}_{easy}$ **do**
$\quad | \quad x_j = x_{\arg \max \cos (x_i, x_j)}$
$\quad | \quad y_i' = \phi(y_i)$
$\quad | \quad y_j' = \phi(y_j)$
$\quad | \quad \tilde{x}_i = \zeta(x_i, x_j)$
$\quad | \quad \tilde{y}_i = \zeta(y_i', y_j')$
**end**
**for** $(x_i, y_i$ **in** $\mathcal{D}_{hard}$ **do**
$\quad | \quad x_j = x_{\arg \max \cos (x_i, x_j)}$
$\quad | \quad y_i' = \phi(y_i)$
$\quad | \quad y_j' = \phi(y_j)$
$\quad | \quad \tilde{x}_i = \zeta(x_i, x_j)$
$\quad | \quad \tilde{y}_i = \zeta(y_i', y_j')$
**end**
**Output:** $\{(\tilde{x}_i, \tilde{y}_i)\}_{i=1}^m$

sample as mentioned in Eq.3. Then, we use the median of the degree of difficulty to partition the dataset: we assign samples with a degree of difficulty less than the median to the easy-to-learn dataset $D_{easy}$, and samples with the degree of difficulty greater than the median to the hard-to-learn dataset $D_{hard}$.

In the second stage of self-evolution learning, we conduct mixup from $D_{easy}$ to $D_{hard}$. For easy-to-learn data, we perform mixup operations on the $D_{easy}$. Given a sample $x_i$ from $D_{easy}$, we search for the most similar sample $x_j$ in $D_{easy}$, where the similarity is measured by cosine similarity. Then, we mix the two samples up by interpolating the inputs ($x_i$ and $x_j$) and labels ($y_i$ and $y_j$) as Eq.1 and Eq.2. The data selected according to the above process is then used for training, and the resulting generated data is added to the model training. In the hard-to-learn dataset, we follow the same way that selects two most similar samples and mixup to

compose a pseudo sample. The sample serves as a new sample to augment the training data. Algorithm 1 summarizes the above procedure.

### 3.4 Instance-Specific Label Smoothing for Regularization

To avoid over-confidence caused by hard labels in few-shot scenarios, we propose a novel instance-specific label smoothing (ILS) approach to adaptively regularize the training and improve the generalization ability of the classification model.

The traditional label smoothing (LS) approach replaces the hard label distribution $y_i$ with $y_i'$ as Eq. 4, where $y_i'$ is a mixture of the original label distribution $y_i$ and a distribution $u_i$. The $u_i$ is usually a uniform distribution.

$$y_i' = (1 - \alpha) * y_i + \alpha u_i \qquad (4)$$

Traditional LS lowers the value of the correct label and increases all others, which successfully prevents the largest predicted score much larger than all others (Szegedy et al., 2016). However, in the traditional LS, the distribution of $u$ is fixed and $u$ cannot dynamically generate labels to adapt to the model learning.

Motivated by this observation, in our instance-specific label smoothing, we propose a sample-aware prior distribution to smooth the labels. Specifically, we replace the fixed distribution $u$ with a dynamic and informative distribution that is adaptively generated by the classification model itself. In practice, similar to Eq. 4, we smooth the label by interpolating the original label $y_i$ with a $p(y|x_i)$ predicted by the classification model. Over all the candidate classes $y_i$ is a one-hot vector, where its value (i.e. probability) on the correct class is 1 and its value on the other class is 0. $p(y|x_i)$ is the model's predicted probability distribution over all the classes. We consider the model prediction $p(y|x_i)$ as the possibility of being the correct label from the model's perspective. As the model is optimized, the model prediction becomes increasingly accurate and the model predicted label approaches the ideal label. We obtain the final smoothed label $y_i'$ as:

$$y_i' = (1 - \alpha) * y_i + \alpha r_i \qquad (5)$$

Then we get the mixed smooth label $\tilde{y}_i'$ through the Eq. 2.

Finally, in the SE training stage, we employ the cross-entropy loss as follows:

$$\mathcal{L}_{LS} = -\frac{1}{m} \sum_{i=1}^{m} \tilde{y}_i' \log p_i \qquad (6)$$

## 4 Experiments

### 4.1 Datasets

To investigate the effectiveness of our method, we conduct extensive experiments on various language understanding tasks, including a diversity of tasks from GLUE (Wang et al., 2018), Super-GLUE (Wang et al., 2019) and other benchmarks, *i.e.*, sentiment analysis (SST-2, Rotten tomato), natural language inference (RTE, CB), paraphrase (MRPC), and text classification (SUBJ, Amazon counterfactual). To simulate the few-shot scenarios, we randomly select 10 samples per class from the training set for each task, and use them for training the models. For evaluation, we use the Accuracy as the metric and report the averaged results over 5 random seeds to avoid stochasticity. Due to the space limitation, we show the details of all tasks and datasets in Appendix A.1 (Table 5).

### 4.2 Implementation Details

We use the representative BERT (Devlin et al., 2019)-BASE and -LARGE models as the backbone PLMs, and fine-tune them in a two-stage manner. Specifically, following many previous mixup methods (Chen et al., 2020; Yoon et al., 2021), we first train the backbone PLMs (without using mixup) with a learning rate of 5e-5, and then continue fine-tuning the models using the mixup strategy with a learning rate of 1e-5. Note that our methods are only adopted in the second stage.

We set a maximum sequence length of 128 and a batch size of 32. AdamW (Loshchilov and Hutter, 2018) optimizer with a weight decay of 1e-4 is used to optimize the model. We use a linear scheduler with a warmup for 10% of the total training step.

### 4.3 Compared methods

We compare our method with other cutting-edge counterparts. Specifically, taking the TMix (Chen et al., 2020) as the base mixup method, we use the following strategies to improve its performance:

- AUM (Park and Caragea, 2022): AUM compares logits to classify samples into two sets and then interpolates samples between these

| Method | SST2 | RTE | MRPC | CB | Rott. | SUBJ | Amazon | Score | |
|---|---|---|---|---|---|---|---|---|---|
| | *Acc.* | *Acc.* | *Acc.* | *Acc.* | *Acc.* | *Acc.* | *Acc.* | *Avg.* | Δ (↑) |
| *Performance of Different No Mixup Methods* | | | | | | | | | |
| BERT-base | 55.43 | 49.59 | 53.67 | 32.49 | 59.46 | 82.72 | 56.46 | 55.68 | – |
| -w/ EDA | 54.72 | 49.80 | 59.20 | 39.60 | 58.96 | 82.06 | 62.2 | 58.07 | +2.39 |
| -w/ Back Translation | 56.85 | 49.20 | 61.60 | 38.94 | 60.67 | 83.16 | 66.63 | 59.57 | +3.89 |
| -w/ CBERT | 54.90 | 49.80 | 57.20 | 32.85 | 60.03 | 82.49 | 62.82 | 57.15 | +1.47 |
| *Performance of Different TMix Improvement Methods* | | | | | | | | | |
| TMix | 54.94 | 49.60 | 61.90 | 41.06 | 56.95 | 83.16 | 58.14 | 57.95 | – |
| -w/ AUM | 56.60 | 49.81 | 62.10 | 42.35 | 58.94 | 83.30 | 65.22 | 59.75 | +1.80 |
| -w/ DMix | 53.68 | 54.40 | 46.40 | 56.80 | 41.80 | 51.76 | 88.66 | 56.21 | -1.74 |
| -w/ SE (Ours) | 57.56 | 49.99 | 62.69 | 42.85 | 58.23 | 83.87 | 68.58 | 60.53 | +2.58 |
| *Performance upon Different Mixup Methods* | | | | | | | | | |
| SSMix | 55.70 | 49.52 | 60.10 | 37.13 | 59.86 | 83.76 | 62.63 | 58.08 | – |
| -w/ SE (Ours) | 56.96 | 49.96 | 61.41 | 39.63 | 61.27 | 84.06 | 65.60 | 59.83 | +1.45 |
| EMbedMix | 53.11 | 49.52 | 61.61 | 37.49 | 58.83 | 83.10 | 63.34 | 58.14 | – |
| -w/ SE (Ours) | 55.89 | 49.88 | 63.28 | 41.07 | 60.10 | 83.86 | 69.22 | 60.46 | +2.32 |
| TreeMix | 55.70 | 49.52 | 60.04 | 37.13 | 59.86 | 83.76 | 62.63 | 58.37 | – |
| -w/ SE (Ours) | 56.96 | 49.96 | 61.17 | 39.63 | 61.27 | 84.06 | 65.60 | 59.80 | +1.43 |

Table 1: Comparison between our SE and the vanilla method applied to mixup methods on the benchmarks. "Rott." is the short for Rotten tomato task. "Δ" denotes the improvement of SE methods compared to the baselines.

| Model | CB | RTE | Rott. | *Avg.* | Δ (↑) |
|---|---|---|---|---|---|
| Baseline | 37.84 | 48.51 | 58.55 | 48.30 | – |
| -w/ SSMix | 42.49 | 48.37 | 59.67 | 50.17 | +1.87 |
| -w/ SE (Ours) | **47.49** | **49.16** | **62.26** | **52.97** | +4.67 |

Table 2: Experimental results of comparison with BERT-large. All values are average accuracy (%) of five runs with different seeds. Models are trained with 10 labeled data per class.

| Learning Strategy | SST2 | Rott. | Amazon | *Avg.* |
|---|---|---|---|---|
| Random | 55.70 | 59.86 | 60.12 | 58.56 |
| Easy-to-hard | **55.81** | **61.17** | **65.37** | **60.78** |
| Hard-to-easy | 55.79 | 61.13 | 64.64 | 60.52 |

Table 3: Experimental results of different data selection methods. All values are average accuracy (%) of five runs with different seeds. Models are trained with 10 labeled data per class.

sets by identifying the most similar and most dissimilar samples from the opposite set.

- DMix (Sawhney et al., 2022): DMix chooses samples based on their diversity in the embedding space.

- SE (Ours): SE divides the dataset into easy-to-learn and hard-to-learn and then mixes up two samples according to the order of difficulty from easy to hard.

Additionally, for reference, we report the results of some traditional data augmentation methods, *i.e.,* EDA (Wei and Zou, 2019) , Back Translation (Shleifer, 2019) and CBERT (Wu et al., 2019). To verify the universality of our SE, we also attempt to adopt it to other base mixup methods, *i.e.,*

EmbedMix (Guo et al., 2019), SSMix (Yoon et al., 2021) and TreeMix (Zhang et al., 2022).

### 4.4 Main Results

The full results of BERT-BASE and -LARGE are shown in Table 1 and Table 2, and we can find that:

**SE surpasses the cutting-edge counterparts in most settings.** When using the TMix as base method, our SE brings much better performance improvements compared to the other counterparts (AUM and DMix), *i.e.*, up to +4.51 average score. Additionally, compared to the other traditional DA methods, SE can also achieve superior performance. These results show the effectiveness of our SE method.

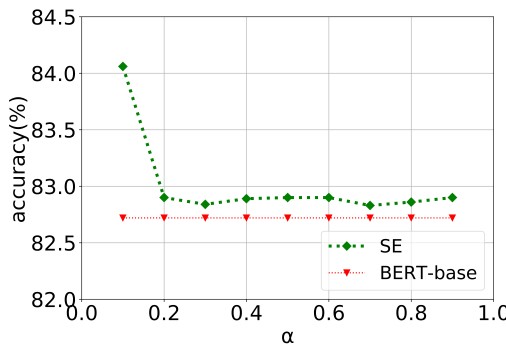

Figure 2: Parameter analysis of $\alpha$ on BERT-base, fine-tuned on SUBJ task.

| Method | SST2 | RTE | Amazon | Avg. | Δ (↑) |
|---|---|---|---|---|---|
| SSMix | 55.81 | 49.73 | 65.37 | 56.97 | – |
| -w/ Vanilla LS | 56.12 | 49.81 | 65.11 | 57.01 | +0.04 |
| -w/ ILS (Ours) | **56.88** | **49.96** | **65.52** | **57.45** | +0.48 |

Table 4: Experimental results of different label smoothing. All values are average accuracy (%) of five runs with different seeds. Models are trained with 10 labeled data per class.

**SE brings consistent and significant performance gains among all baselines.** In addition to the TMix, we also adopt our SE to more base mixup methods, *i.e.*, SSMix, EmbedMix and TreeMix, and show the contrastive results in Table 1. As seen, compared to the baselines, our SE can bring consistent and significant performance gains among all these methods, indicating its universality.

**SE works well in both model sizes.** Here, we verify whether our SE can still work in the large model scenarios. Taking some tasks as examples, we show the contrastive results in Table 2. It can be seen that, with the help of our SE, BERT-large achieves much better performance against the baselines. These results prove the effectiveness of our SE in both model sizes.

### 4.5 Ablation Studies

We evaluate the impact of each component of our SE, including *i*) learning strategy on mixup, *ii*) instance-specific label smoothing approach, *iii*) coefficient $\alpha$.

**Impact of Learning Strategy on Mixup.** As mentioned in §3.3, we perform the mixup process in an easy-to-hard manner, *i.e.*, first mixing the easy samples and then mixing the hard samples. Here, to investigate the impact of different learning strategies on mixup, we conduct contrastive exper-

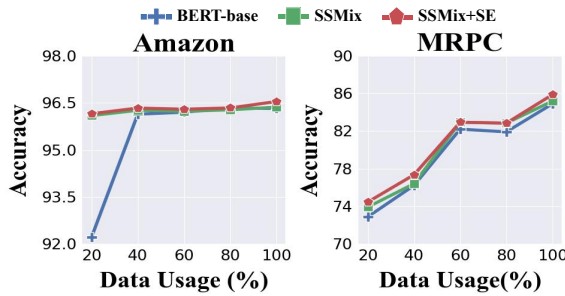

Figure 3: Results at various training data sampling rates. BERT-base models fine-tuned on Amazon and MRPC are used. We can see that our method achieves better performance across all data size regimes, especially in the few-shot scenarios.

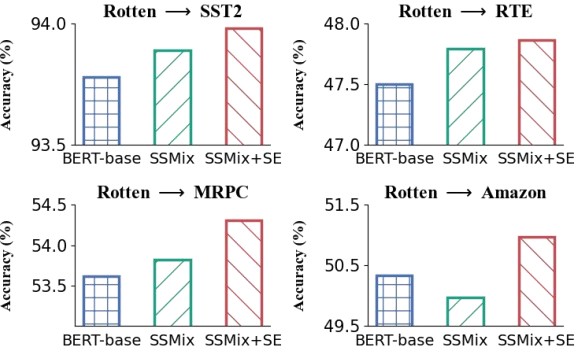

Figure 4: Analysis of task generalization. The model is fine-tuned on the Rotten tomato task and transferred to four different tasks. We can see that our SE method consistently brings better generalization compared with its counterparts.

iments as following: 1) "Random": we randomly select the samples from full dataset; 2) "Easy-to-hard": we first train the model with easy samples and then with hard samples; 2) "Hard-to-easy": the opposite learning order to "Easy-to-hard". The detailed results are listed in Table 3, and we can find that both ordered learning strategies outperform the baseline "Random", indicating the significance of evolution learning. More specifically, "Easy-to-hard" achieves the best performance, thus leaving it as the default setting.

**Comparison of Different Label Smoothing.** A key technology in our method is the instance-specific label smoothing method. To verify its effectiveness, we compare it with vanilla label smoothing and report the results in Table 4. We show that 1) both label smoothing methods achieve better performance compared to the baseline, confirming the necessity to alleviate the over-

confidence problem; 2) our method could further improve the results by a clear margin against vanilla label smoothing. These results prove the effectiveness of our ILS method.

**Impact of Coefficient $\alpha$.** The weight $\alpha$ in Eq. 4 is used to control the ratio of label smoothing, which is an important hyper-parameter. In this part, we examine its impact by evaluating the performance with different $\alpha$ on SUBJ task, and illustrate the results in Figure 2. As shown, compared with the baseline, our method consistently achieves better performance across all ratios of $\alpha$. More specifically, the case of $\alpha = 0.1$ performs best, and we hereby use the setting in our experiments.

### 4.6 Expanding to High-resource Scenarios

Although our work mainly focuses on the data augmentation in few-shot tasks, we also investigate whether our method still works in the high-resource scenarios. Specifically, we change the percentage of training data used from 20% to 100% and illustrate the results of several tasks in Figure 3.

As expected, our method achieves significant performance improvements when the amount of training data was extremely limited, continuing to confirm the effectiveness of our method. Moreover, we can also observe performance gains brought by our SE in the other relatively high-resource scenarios. These results prove the universality of our method.

### 4.7 Analysis of Model Generalization

To investigate whether our SE can bring better model generalization, we conduct experiments from two perspectives: *i*) measuring the cross-task zero-shot performance, and *ii*) visualizing the loss landscapes of models.

**Task Generalization.** The performance of out-of-domain (OOD) data is widely used to verify the model generalization (Xu et al., 2021; Zhong et al., 2022). Hence, we follow Zhong et al. (2022) and and evaluate the performance of models on several OOD data. In practice, we first fine-tune BERT-based models trained with different methods (including "Baseline", "SSMiX", and "SSMix+SE") on the Rotten Tomato task, and then inference on other tasks, *i.e.*, SST2, MRPC, RTE, and Amazon. The results are illustrated in Figure 4. We observe that "SSMix+SE" consistently outperforms the other counterparts. To be more specific, compared with baseline, our SE brings a +0.47 average improvement score on these tasks, indicating that our method boosts the performance of models on OOD data.

**Visualization of Loss Landscape.** To have a closer look, we also visualize the loss landscapes of different BERT-base models fine-tuned on the Rotten Tomato task. In practice, we follow the "filter normalized" setting in Li et al. (2018) and show the 3D loss surface results in Figure 5. We can see that our method has flatter smoother surfaces compared to others. This result proves that SE can smooth the loss landscape and improve the generalization of models effectively.

## 5 Conclusion

In this paper, we propose a simple-yet-effective self-evolution (SE) learning mechanism to improve the existing mixup methods on text classification tasks. SE for mixup follows two stages: conducting data division based on the degree of difficulty and mixup based on the order from easy to hard. SE can be used in various mixup methods to generate more adaptive and model-friendly pseudo samples for model training. Also, to avoid over-confidence in the model, we propose a novel instance-specific label smoothing approach. Extensive experiments on four popular mixup methods, EmbedMix, TMix, SSMix, and TreeMix, verify the effectiveness of our method. Quantitative analyses and in-depth discussions show our method improves the generalization, and robustness of models.

## Limitations

Our work has several potential limitations. First, due to limited computational resources, we only validate our self-evolution learning on base- and large-size BERT models. Expanding our experiments to larger model sizes would make our work more convincing. On the other hand, for the results of baseline methods, we should compare our results with those in the original paper for a fair comparison. However, due to the difference of PLMs and tasks used in the other baselines and ours, it is unreasonable to compare the results directly. Hence, as an alternative, we only reproduce the results in our settings using the code in the corresponding papers.

## Ethics Statement

We take ethical considerations very seriously, and strictly adhere to the EMNLP Ethics Policy. This paper proposes a self-evolution learning algorithm

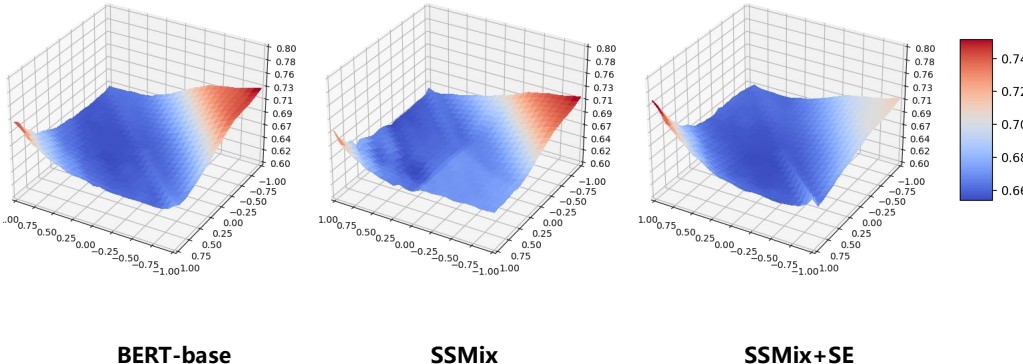

**BERT-base**      **SSMix**      **SSMix+SE**

Figure 5: The 3D loss surface comparison between baseline, vanilla SSMix, and our SE methods applied to BERT-base. Note that the PLMs are fine-tuned on the Rotten tomato task. It can be seen that SE methods significantly smooth the loss surface, i.e., improving the model generalization effectively.

to improve the existing mixup strategy. The proposed approach aims to precisely augment the few-shot training data with the original training corpus, instead of encouraging the model to generate new sentences that may cause the ethical problem. Moreover, all pre-trained language models and downstream datasets used in this paper are publicly available and have been widely adopted by researchers. Thus, we believe that this research will not pose ethical issues.

## Acknowledgements

This work is supported by the following foundations: the National Natural Science Foundation of China under Grant No.62025208 and No.62306330, the Xiangjiang Laboratory Foundation under Grant No.22XJ01012.

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

# A  Appendix

## A.1  Details of Datasets

| Dataset | Task | # Label | Size |
|---|---|---|---|
| SST-2 | Sentiment | 2 | 67k / 1.8k |
| RTE | NLI | 2 | 2.5k / 3k |
| MRPC | Paraphrase | 2 | 3.7k / 1.7k |
| CB | NLI | 3 | 556 / 250 |
| SUBJ | Classification | 2 | 8k / 2k |
| Rotten tomato | Sentiment | 2 | 8.53k / 1.07k |
| Amazon counterfactual | Classification | 2 | 5k / 5k |

Table 5: Dataset name, task, number of total labels, and dataset size of datasets we used as a benchmark. The task column describes the objective of each dataset.

## A.2  Parameter Analysis on $\lambda$

As stated in Sec3.2, We use the parameter $\lambda$ to control the mixing ratio of two samples. Here, we analyze the influence of different $\lambda$ in detail. In practice, we used TMix as the baseline and conducted experiments with $\lambda$ varying from 0.1 to 0.4. The results are presented in the table 6. As seen, our method brings consistent performance gains across various, indicating that our method is not sensitive to the value of $\lambda$. Notably, "$\lambda$=0.2" achieves the best performance, thus leaving as the default setting.

| $\lambda$ | 0.1 | 0.2 | 0.3 | 0.4 |
|---|---|---|---|---|
| TMix | 55.01 | 54.94 | 54.46 | 54.93 |
| TMix+Ours. | 55.66(+0.65) | 57.56(+2.62) | 54.93(+0.47) | 55.40(+0.47) |

Table 6: The experimental results were an average of 5 runs with random seeds. The content in the brackets is the improvement of our method.