# OpenReview forum: "Self-Evolution Learning for Mixup: Enhance Data Augmentation on Few-Shot Text Classification Tasks"
_EMNLP/2023/Conference — EMNLP 2023 Main_

### Official Review · Reviewer_RCEN · 2023-07-27

**Soundness:** 3

**Excitement:**

3: Ambivalent: It has merits (e.g., it reports state-of-the-art results, the idea is nice), but there are key weaknesses (e.g., it describes incremental work), and it can significantly benefit from another round of revision. However, I won't object to accepting it if my co-reviewers champion it.

**Paper Topic And Main Contributions:**

This paper introduces a mixup-based data augmentation method for improving few-shot text classification with pretrained language models. Specifically, the authors leverage a curriculum strategy to effectively employ training data and propose a label smoothing method that can fit the curriculum learning strategy.

**Reasons To Accept:**

1. The instance-specific label smoothing method is straightforward and effective.
2. Better experimental results than other mixup counterparts.

**Reasons To Reject:**

1. Lack comparison with other fine-tuning methods.
2. The overconfidence issue is a common problem of overparameterized models like pretrained language models, to what extent this problem can be partially solved by label smoothing is not shown in this paper. Since this is the key hypothesis of this paper, experimental analysis about the prediction logits distribution with/without the proposed label-smoothing method should be available.
3. Line 51 - 66 introduces data augmentation from the angle of changing to the label, it makes this paragraph hard to convey information to audiences who are not familiar with data augmentation, though it greatly fits the topic of this paper. A better way might be DA methods based on synthesis such as mixup and others like back-translation and EDA.
4. The figure 1 is hard to read and adding a figure to display how equation (5) evolves, by referring to models performance in test data, would be much more inspiring since audiences would spend more time on understanding the content of section 3.4.
5. What makes us to link loss surface smoothness in figure 5 and the propose method?

**Reproducibility:**

4: Could mostly reproduce the results, but there may be some variation because of sample variance or minor variations in their interpretation of the protocol or method.

**Reviewer Confidence:**

5: Positive that my evaluation is correct. I read the paper very carefully and I am very familiar with related work.

---

> ### Author Rebuttal · Authors · 2023-08-28
>
> Thank you for your thoughtful review! Here we address your questions about the work.
>
> ### R1: Lack comparison with other fine-tuning method.
>
> We will compare some other methods, but due to time constraints, now we only compared Text smoothing [1]The comparison results are shown in the table below.
>
> | Method         | TREC           |
> |----------------|----------------|
> | Text Smoothing | 67.51%         |
> | Ours           | 68.13%(+0.62%) |
>
> Based on the results presented in the table, it can be observed that our method has an advantage over the other methods compared.
>
> ### R2: extent over-confidence can be partially solved by label smoothing is not shown in this paper
>
> Label smoothing **breaks the model's excessive dependence on specific labels** by introducing uncertain labels during training. Each sample's label is modified into a probability distribution instead of a rigid 0 or 1. For example, in a binary classification problem, the label may be set as 0.9 (positive class) and 0.1 (negative class), instead of simply 0 or 1. This way, the model needs to learn from a probability distribution instead of just learning a hard label. Intuitively, **estimating a distribution can be seen as studying from the target with a confidence** score (the probability acts as the confidence score ), but estimating a hard label can be seen as learning from a target with 100% confidence, thus leading to over-confidence.
>
> We **conducted an experiment** to calculate the statistical loss for the samples with and **without instance-specific label smoothing (ILS). And we compared their corresponding variances. The results are shown in the following table, which demonstrates that the sample loss with ILS is more stable.
>
>
>
> |       | variance |
> |-------|----------|
> | without ILS | 0.039    |
> | with ILS    | 0.013    |
>
> ### R3:  the introductions of line51-61 makes this paragraph hard to convey information to audiences who are not familiar with data augmentation,
>
> We will describe the DA method in a more accessible way, as described below.: DA methods based on synthesis such as mixup and others like back-translation and EDA.
>
>
> ### R4: The Figure 1 is hard to read and adding a figure to display how equation (5) evolves,
>
> We commit to providing the changes in the next version. In the rebuttal stage, we give the idea of the figure.  We use **a test sample**: "take on loss and loneliness"  , One-hot label [1 0] of this sample is transformed into [0.73 0.27] after label smoothing. We use a **histogram to represent the label distribution**. This allows us to directly observe the changes in the labels. Additionally, we will show the corresponding changes in the sample's loss.
>
>
>
> ### R5: What makes us link loss surface smoothness in Figure 5 and the proposed method?
>
> The smoothness of the loss surface can intuitively reflect the **generalization performance** of a model [2,3]and reflect the over-confidence issue is relieved. When the loss surface is smooth, it indicates that the model can adapt well to small perturbations in the input, and the loss value does not change significantly.
>
> We mentioned before that our approach would cause an over-confidence problem, so we tried to introduce an instance-specific label smoothing method. And now we have proven that our approach can indeed improve generalization (smoother loss landscape and better ood effect), confirming our claim.
>
>
> [1] Text Smoothing: Enhance Various Data Augmentation Methods on Text Classification Tasks] (Wu et al., ACL 2022).
>
> [2] Visualizing the Loss Landscape of Neural Nets
>
> [3] Sharpness-Aware Minimization for Efficiently Improving Generalization

---

### Official Review · Reviewer_YbFy · 2023-08-05

**Soundness:** 2

**Excitement:**

3: Ambivalent: It has merits (e.g., it reports state-of-the-art results, the idea is nice), but there are key weaknesses (e.g., it describes incremental work), and it can significantly benefit from another round of revision. However, I won't object to accepting it if my co-reviewers champion it.

**Paper Topic And Main Contributions:**

This paper proposes a self-evolution learning-based mixup data augmentation method to address the problem of data scarcity in few-shot text classification scenarios.

The method is primarily divided into two stages: conducting data division based on the degree of difficulty and performing mixup in the order from easy to hard.To further alleviate the model over-confidence, this paper also proposes an instance-specific label smoothing regularization approach which linearly interpolates  the model's output and one-hot labels of the original samples to generate new soft labels for label mixing up.

**Questions For The Authors:**

1. The improvements observed in the ablation experiments are marginal, and the significance of these results is unclear. Can you perform statistical tests, such as a t-test, to validate the effectiveness of the proposed "instance-specific label smoothing" method? How do you respond to the observed minimal differences in accuracy, and what do they indicate about the practical applicability of your approach?

2. Your experimental design appears limited in scope, particularly concerning the analysis of hyperparameters and the range of models used for comparison. Could you provide more insights into the choice of hyperparameters and why some were not analyzed experimentally? Would you consider conducting experiments with other models such as Bart, T5, RoBERTa, to demonstrate the generality of your proposed method, and how do you envision these additional experiments strengthening or altering the conclusions of the paper?

**Reasons To Accept:**

1.The paper proposes a simple and explainable data augmentation method.

2.The diagrams illustrating the model structure are clear and concise.

3.The writing is overall good.

**Reasons To Reject:**

1.This paper appears to simply divide the dataset based on the learning difficulty of the samples and apply the mixup methods, without presenting any particularly innovative aspects. I think that the innovation of this paper is limited.

2.The credibility of the ablation experiments in this paper is insufficient. Among them, the ablation experiments for "different data selection methods" show that the accuracy difference between Easy-to-hard and Hard-to-easy is only 0.2%. However, the ablation experiments for the proposed "instance-specific label smoothing" method demonstrate a marginal improvement of 0.48% (less than 1%) compared to the baseline, which does not sufficiently indicate the effectiveness of this method and requires a t-test for further validation.

3.There are some errors in the paper, mainly concentrated on the fourth page of the paper.

1)In the sentence "we assign samples with a degree of difficulty less than the median to the hard-to-learn dataset ", "hard-to-learn" should be modified to "easy-to-learn".

2) In the sentence ",The sample serves as a new sample to augment the training data.", the punctuation before "The" should be modified to a period.

3) The sentence "we mix the two samples up by interpolating the inputs ( and ) and labels ( and ) as Eq.1 and Eq.2. The data selected according to the above process is then used for the mixup" is repetitive and not fluent, and needs to be revised.

4. The paper lacks some experiments. Regarding hyperparameters, this paper only conducts experimental analysis on the parameter α of instance-specific label smoothing, while the hyperparameter λ of the mixup method is not analyzed experimentally. Additionally, this paper only compares with BERT-base and BERT-large in the experimental setup. I believe it is necessary to conduct experiments with more models, such as Bart, T5, RoBERTa and others, to validate the generality of the proposed method in this paper.

**Reproducibility:**

3: Could reproduce the results with some difficulty. The settings of parameters are underspecified or subjectively determined; the training/evaluation data are not widely available.

**Reviewer Confidence:**

4: Quite sure. I tried to check the important points carefully. It's unlikely, though conceivable, that I missed something that should affect my ratings.

---

> ### Author Rebuttal · Authors · 2023-08-28
>
> Motivated by your comments, we have deeply reconsidered the architecture of our work and tried to fix all the problems you mentioned.
>
> ### R1: The innovation of this paper is limited.
>
> As stated Our work not only proposes a **new mixup training strategy** but also proposes a **dynamic label smooth** method: instance-specific label smooth.  Our proposed method caters to the **growth of the model learning ability** and adapts to the ability when generating training samples.
>
> Moreover, compared to other mixup methods, the method we propose is not customized for specific scenarios and have the potential to be applied to various mixup methods for improvement.
>
>
> ### R2:· The accuracy difference between Easy-to-hard and Hard-to-easy is only 0.2% and the "instance-specific label smoothing" method demonstrates a marginal improvement of 0.48%
>
> Compared with the mixup method proposed in other papers, the average improvement of the baseline, SSMix [1] has an average improvement of about 0.4% compared with the baseline in the GLUE dataset and the Mixup-transformer has an average improvement of about 0.56% in line 8 of Table 1 in their paper. The improvement of 0.48% (line 4 in Table 2) achieved by the proposed label-smooth method is higher than 0.4%. Moreover, the interpolation techniques of 0.2% Eassy-to-hard and hard-to-easy (Table 3) are in the same order of magnitude as 0.4% and 0.56%.  Therefore, the improvement margin of our method is sufficient compared to other papers in this field.
>
>
> ### R3: some errors in the sentence
>
> Thank you for your detailed inspection and we apologize for our negligence. We have made revisions to these errors in the paper as follows:
>
> (1)In line 275,  we have modified "hard-to-learn"  to "easy-to-learn".
>
> (2)In line 293,  we have modified the punctuation before 'The'.
>
> (3)In line 286, we re-write the influent sentence as "We mix the two samples by interpolating the inputs and labels as Eq.1 and Eq.2. The data selected according to the above process is then used for training".
>
> ### R4: the hyperparameter $\lambda$ of the mixup method is not analyzed experimentally.
>
> For your suggested experimental analysis of the λ parameter, at the rebuttal stage, we have conducted the experiment and analyzed the effectiveness of our method with **different values of λ**. We used TMix as the baseline and conducted experiments with λ varying from 0.1 to 0.4. The experimental results were an average of 5 runs with random seeds. The results are presented in the following table. The content in the brackets is the improvement of our method.  The content in the brackets is the improvement of our method.
>
>
> | $\lambda$         | 0.1          | 0.2          | 0.3          | 0.4          |
> |------------|--------------|--------------|--------------|--------------|
> | TMix       | 55.01        | 54.94        | 54.46        | 54.93        |
> | TMix+Ours. | 55.66(+0.65) | 57.56(+2.62) | 54.93(+0.47) | 55.40(+0.47) |
>
> As seen, our method brings consistent performance gains across various $\lambda$, indicating that our method is not sensitive to the value of $\lambda$. Notably, "λ=0.2" achieves the best performance, thus leaving as the default setting.
>
> ### R5: Can you perform statistical tests to validate the effectiveness of the proposed "instance-specific label smoothing" method? and  respond to the observed minimal differences in accuracy
>
> Sure, for the statistical tests, we follow the suggestion of reviewer 2 (g5MQ): conducting each experiment five times and calculating the mean and variance of the results. The results are shown in the table below.
>
> |                 | SST2   | RTE    | Amazon |
> |-----------------|--------|--------|--------|
> | SSMix(baseline) | 55.81% | 49.73% | 65.37% |
> | SSMix+ILS(ours) | 56.88% | 49.96% | 65.52% |
> | Difference      | +1.07% | +0.23% | +0.18% |
> | Varience        | 0.31%  | 0.03%  | 0.02%  |
>
> From the variance in the table, it can be seen that the ILS we proposed has a stable improvement and is effective.
> For the issue of observed minimal differences in accuracy, please refer to the response to R2.
>
> ### R6: Provide more insights into the choice of hyperparameters, why some were not analyzed experimentally
> and would you consider conducting experiments with other models such as Bart, T5, RoBERTa
>
> Before submission, we conducted experiments and found that the best performance was obtained when a was set to 0.1. The following table presents the results obtained in that experiment.
>
> | α    | 0.1   | 0.2  | 0.3   | 0.4   | 0.5  | 0.6  | 0.7   | 0.8   | 0.9  |
> |------|-------|------|-------|-------|------|------|-------|-------|------|
> | Οurs | 84.06 | 82.9 | 82.84 | 82.89 | 82.9 | 82.9 | 82.83 | 82.86 | 82.9 |
>
> For mixup, we used the mixing proportion settings as described in the original mixup papers [1,2,3,4].  To avoid unfair comparison due to tuning parameters, we used the default hyperparameter values consistent with the baseline.  Now we have conducted experiments on the mixup hyperparameter λ, and the results are shown in the table below.
>
> | λ          | 0.1          | 0.2          | 0.3          | 0.4          |
> |------------|--------------|--------------|--------------|--------------|
> | TMix       | 55.01        | 54.94        | 54.46        | 54.93        |
> | TMix+Ours. | 55.66(+0.65) | 57.56(+2.62) | 54.93(+0.47) | 55.40(+0.47) |
>
> Additionally, for hidden-level mixup methods such as TMix, the number of mixed layers was also set to the default values.We will add more parameter analyses and experimental details in the revised paper.
>
> Sure, we are willing to conduct experiments on other pre-trained models. Due to time constraints, we did not conduct these experiments during the rebuttal stage, but we will verify our method on the T5 and Roberta models and include the results in our paper.
>
>
> [1] SSMix: Saliency-Based Span Mixup for Text Classification (Yoon et al., ACLFindings 2021)
>
> [2] Mixup-Transformer: Dynamic Data Augmentation for NLP Tasks.(Sun, Lichao, et al., COLING2020)
>
> [3] DMix: Adaptive Distance-aware Interpolative Mixup (Sawhney et al., ACL 2022)
>
> [4]MixText: Linguistically-Informed Interpolation of Hidden Space for Semi-Supervised Text Classification (Chen et al., ACL 2020)

---

### Official Review · Reviewer_g5MQ · 2023-08-11

**Soundness:** 3

**Excitement:**

4: Strong: This paper deepens the understanding of some phenomenon or lowers the barriers to an existing research direction.

**Missing References:**

Sahu, G., et al. "Data augmentation for intent classification with off-the-shelf large language models. CoRR (2022)."

**Paper Topic And Main Contributions:**

This paper presents a method called self-evolution learning (SE) for better mixup data augmentation in text classification tasks. The authors split the text data into two sets: easy and difficult samples with the help of BERT. They then apply a method called mixup, starting with the easy set and gradually moving to the difficult ones. To prevent the model from being overconfident in its predictions, they use a method to create soft labels based on the original hard labels. This approach enhances the model's ability to handle different types of text and perform better on text classification tasks, as demonstrated through extensive experiments.

**Reasons To Accept:**

- The paper is well written and easy to read, I like Figure 1in the way it shows the pipeline of the method and Algorithm 1 in the way it shows more specific control flow of the data.

- The authors did a great job at testing their SE method on a wide variety of benchmarks showing consistent improvements and illustrating that that the division of the samples to easy/difficult splits is a promising strategy that goes well with mixup.

- The authors made extensive ablation studies that shows the efficacy of the proposed method and the robustness of the hyperparameters such as the alpha of BERT-Base



**Reasons To Reject:**

- I encourage the authors to make the best results in bold to make the experiment tables easier to parse.

- I encourage the authors to run each experiment 5 times and report the mean and variance to see the significance of the records

- No code was provided to help in checking the specific details of the pipeline and reproduce the results

- It looks like from Table 3 that hard-to-easy achieves very similar results to easy-to-hard. This result makes me wonder whether the "easy"/"hard" strategy is what makes this method outperform others or whether there are additional components that this method has that makes it SOTA. This phenomenon needs to be explained.

**Reproducibility:**

4: Could mostly reproduce the results, but there may be some variation because of sample variance or minor variations in their interpretation of the protocol or method.

**Reviewer Confidence:**

3: Pretty sure, but there's a chance I missed something. Although I have a good feel for this area in general, I did not carefully check the paper's details, e.g., the math, experimental design, or novelty.

---

> ### Author Rebuttal · Authors · 2023-08-28
>
> Thank you for your thoughtful and positive comments! We follow your suggestions and we believe we have solved all your concerns as follows!
>
> ### R1: make the best results in bold
>
> We will follow your suggestions to bold the results in our next version.
>
> ### R2: run each experiment for 5 times and report means and variation
>
> Thanks! In fact, the results reported in our paper were already the mean is the average of five random seed runs (mentioned in **line 357** of the paper), and we will include the variation results in the next version of the paper. The partial results are shown in the table below.
>
> | **Method**          | **SST2**   | **RTE**    | **MRPC**   |
> |-----------------|--------|--------|--------|
> | SSMix(baseline) | 55.70% | 49.52% | 60.10% |
> | Ours            | 56.96% | 49.96% | 61.41% |
> | Difference      | +1.26% | +0.44% | +1.31% |
> | Varience        | 0.47%  | 0.15%  | 0.69%  |
>
> ### R3: no code was provided to help in checking the specific details
>
> We have promised that we would release our code after the paper's acception.  At the rebuttal stage, we **release our code** in an anonymous link: https://anonymous.4open.science/r/Self-evolution-learning-for-mixup-4F8D/
>
> ### R4: hard-to-easy achieves very similar results to easy-to-hard
>
> Regarding the issue of the gap between "easy-to-hard" and "hard-to-easy" in Table 3, we can refer to the average improvement effect of the mixup method in other papers.
>
> The average accuracy improvement of the Mixup-Transformer [1]  method in line 8  of Table 1 in their paper is 0. 56% and the SSMix [2]  method has an average improvement effect of 0.4% on the GLUE dataset compared to the baseline, and the difference between  "easy-to-hard" and "hard-to-easy"  is 0.2%, which is in the same order of magnitude. Therefore, the improvement margin of our method is sufficient enough to distinguish two of our variants in ablation studies.
>
> [1] Mixup-Transformer: Dynamic Data Augmentation for NLP Tasks.(Sun, Lichao, et al., COLING2020)
>
> [2] SSMix: Saliency-Based Span Mixup for Text Classification (Yoon et al., ACLFindings 2021)
>
> ### R5：missing References: Sahu, G., et al. "Data augmentation for intent classification with off-the-shelf large language models. CoRR (2022)."
> We fully agree with your comment regarding the missing reference, and we will include it in our paper.

---

### Official Review · Reviewer_hra8 · 2023-08-12

**Soundness:** 4

**Excitement:**

3: Ambivalent: It has merits (e.g., it reports state-of-the-art results, the idea is nice), but there are key weaknesses (e.g., it describes incremental work), and it can significantly benefit from another round of revision. However, I won't object to accepting it if my co-reviewers champion it.

**Paper Topic And Main Contributions:**

This paper proposes a new approach to data augmentation in text classification called Self-Evolution Learning for Mixup. The main problem addressed in this paper is the over-confidence problem in few-shot text classification tasks, where the model tends to overfit to the limited training data and perform poorly on unseen data.
The main contributions of this paper are:
1. A new data augmentation approach called Self-Evolution Learning for Mixup that generates more adaptive and model-friendly pseudo samples.
2. An instance-specific label smoothing regularization approach that alleviates the over-confidence problem in few-shot text classification tasks.
3. A comprehensive evaluation of the proposed method on various benchmark datasets, demonstrating its effectiveness and superiority over existing methods.
Overall, this paper provides a novel and effective solution to the over-confidence problem in few-shot text classification tasks, which has important implications for real-world applications of natural language processing.

**Reasons To Accept:**

The strengths of this paper include its novel approach to data augmentation in text classification, its comprehensive evaluation on various benchmark datasets, and its instance-specific label smoothing regularization approach that alleviates the over-confidence problem in few-shot text classification tasks. This paper has the potential to make a significant contribution to the NLP community by addressing an important problem in few-shot text classification and providing a novel and effective solution that can be applied to other NLP tasks as well.

**Reasons To Reject:**

For the results of baseline methods, the authors didn't directly compare the results with those in the original paper, due to the difference of PLMs and tasks used in the other baselines and ours. As an alternative, the authors only reproduce the results in their settings using the code in the corresponding papers. There may remain a risk on validating the true contribution/improvement of this approach.

**Reproducibility:**

4: Could mostly reproduce the results, but there may be some variation because of sample variance or minor variations in their interpretation of the protocol or method.

**Reviewer Confidence:**

3: Pretty sure, but there's a chance I missed something. Although I have a good feel for this area in general, I did not carefully check the paper's details, e.g., the math, experimental design, or novelty.

---

> ### Author Rebuttal · Authors · 2023-08-28
>
> Thank you for your constructive and valuable comments! Following your suggestions, we are happy to clarify our concerns and refine our paper.
>
> ### R1:  didn't directly compare the **results with those in the original paper**
>
> In this paper, we primarily consider the problem of data augmentation **in the few-shot setting**. Therefore, we designed extensive experiments in few-shot scenarios to validate the performance of the proposed method. However, the results in the original paper are mainly based on the **full dataset (instead of the few-shot setting)**.
>
> We also recognize the relevance of comparing to the original dataset. In Figure 3 of our paper submission, we investigated the performance of our method with different amounts of data, including a comparison of model performance on some datasets with full data sets.
> We have released our code, which ensures other researchers can reproduce our results in the few-shot setting.

---

### Meta-Review · Area_Chair_CmZx · 2023-09-19

**Recommendation:** 4

**Metareview:**

This work studies mixup for few-shot text classification and proposes a 2-stage adaptation using two different subsets of the data. Reviewers in general agree about the experimental rigor of this work and found it novel and well-written. There are some concerns about the limited improvements observed. Though a SOTA result with big improvement is not a requirement for acceptance, it reduces the excitement felt by the reviewers.

---

### Decision · Program_Chairs · 2023-10-07

**Decision:**

Accept-Main

**Comment:**

This work studies mixup for few-shot text classification and proposes a 2-stage adaptation using two different subsets of the data. Reviewers in general agree about the experimental rigor of this work and found it novel and well-written. There are some concerns about the limited improvements observed. Though a SOTA result with big improvement is not a requirement for acceptance, it reduces the excitement felt by the reviewers.